# Analysis of Failed Two-Stage Procedures with Resection Arthroplasty as the First Stage in Periprosthetic Hip Joint Infections

**DOI:** 10.3390/jcm10215180

**Published:** 2021-11-05

**Authors:** Sebastian Simon, Bernhard J.H. Frank, Susana Gardete, Alexander Aichmair, Jennyfer Angel Mitterer, Martin Dominkus, Jochen G. Hofstaetter

**Affiliations:** 1Michael-Ogon Laboratory for Orthopaedic Research, Orthopaedic Hospital Vienna-Speising, 1130 Vienna, Austria; sebastian.simon@oss.at (S.S.); bernhard.frank@oss.at (B.J.H.F.); Susana.hartman@oss.at (S.G.); alexander.aichmair@oss.at (A.A.); JennyferAngel.Mitterer@oss.at (J.A.M.); 2II. Department of Orthopaedic Surgery, Orthopaedic Hospital Vienna-Speising, 1130 Vienaa, Austria; martin.dominkus@oss.at; 3School of Medicine, Sigmund Freud University, 1020 Vienna, Austria

**Keywords:** resection arthroplasty, two-stage procedure, risk-factors, PJI

## Abstract

Resection arthroplasty can be performed as the first stage of a two-stage procedure in some patients with severe periprosthetic hip joint infections with poor bone stock. This retrospective study aimed to evaluate factors associated with the subsequent failure or success of these patients. Between 2011 and 2020; in 61 (26.4%) of 231 patients who underwent a two-stage protocol of periprosthetic hip joint infections; no spacer was used in the first stage. The minimum follow-up period was 12 months. Patient’s demographics and various infection risk factors were analyzed. In total, 37/61 (60.7%) patients underwent a successful reimplantation, and four patients died within the follow-up period. Patients within the failure group had a significantly higher Charlson comorbidity index (*p* = 0.002); number of operations prior to resection arthroplasty (*p* = 0.022) and were older (*p* = 0.018). Failure was also associated with the presence of a positive culture in the first- and second-stage procedures (*p* = 0.012). Additional risk factors were persistent high postoperative CRP values and the requirement of a negative-pressure wound therapy (*p* ≤ 0.05). In conclusion, multiple factors need to be evaluated when trying to predict the outcome of patients undergoing resection arthroplasty as the first stage of a two-stage procedure in patients with challenging periprosthetic hip joint infections.

## 1. Introduction

Two-stage exchange arthroplasty with a temporary antibiotic-loaded cement spacer remains the most widely used strategy for the management of chronic periprosthetic hip joint infections [1,2]. In severe cases with poor bone stock and/or abductor insufficiency, a non-spacer option can be performed [3]. However, this resection arthroplasty procedure generally leads to a poor functional status in the interim stage [4,5]. Therefore, a better understanding and identification of the predictive risk factors associated with resection arthroplasty failure are crucial to improve clinical and surgical management [4,5].

Patients’ risk factors associated with periprosthetic joint infections (PJIs) include surgical site infection score (SSI), Charlson comorbidity index (CCI), BMI, serum C-reactive protein (CRP) levels, older age and sex [6,7,8,9,10,11,12,13]. However, these factors are not always associated with persistent infections or failure of a septic revision surgery. Ineffective two-stage exchange arthroplasties are related to unsuccessful eradication of the original infection during reimplantation [14,15,16,17]. Occurrence of polymicrobial surgical infections and the presence of multidrug-resistant pathogens play a significant role in the high resection arthroplasty failure rate [18,19] and substantial increased numbers of readmission and reoperations [11,20,21]. Wound leakage is a sign of persistent infection and requires repeat debridement [22]. Negative-pressure wound therapy (NPWT) could be used in medically unfit patients, but its effectiveness is unknown [12,23,24,25,26]. In general, outcome studies in patient groups with PJIs are hard to compare due to their heterogeneous characteristics. For the management of PJIs and to predict the outcome of PJIs, all potential risk factors should be considered and evaluated [3]. There are several reported studies about factors associated with the treatment outcome in patients with spacer implantation as the first-stage procedure [17,26]. However, there is limited literature available regarding possible risk factors associated with failed resection arthroplasties as a first-stage procedure.

The aim of this retrospective study was to identify predictive risk factors associated with the outcome of resection arthroplasties as a first stage in a two-stage procedure. Comorbidities, microbiological spectrum, serum values and prolonged wound leakage were evaluated using propensity score matching (PS) between failed and successful resection arthroplasties as the first stage of a two-stage procedure.

## 2. Materials and Methods

### 2.1. Patient Data Acquisition

After the institutional review board approval, we performed a retrospective, cross-sectional single-center analysis using our institutional arthroplasty registry as well as our prospectively maintained PJI infection database. Between January 2011 and December 2020 a total of 1684 revision hip arthroplasty procedures were performed. There were 490/1684 (29.1%) revisions due to PJIs. In total, 231/490 (47.1%) patients underwent a two-stage procedure: 170/231 (73.6%) had a spacer implantation and 61/231 (26.4%) were submitted to a resection arthroplasty intervention as the first stage of a two-stage procedure. Institutional guidelines recommend a spacer implantation as the first stage. A resection arthroplasty was only performed in patients with poor bone stock and a difficult soft tissue situation based on the clinicians’ judgment. During resection arthroplasties, all prosthetic components were removed without spacer implantation. All 61 resection arthroplasties were included in this study.

### 2.2. Patient Demographics and Factors Associated with a Failed Two-Stage Protocol

We analyzed patient demographics: body-mass-index (BMI), age and sex. Other infection risk factors, such as surgical side infections (SSI, range: 0–35), Charlson comorbidity index (CCI; range: 0–37) and the number of operations prior to and after resection arthroplasty procedure, were compared [27,28]. We further examined the re-infection and re-revision rates and assessed both the microbiological spectrum and antimicrobial resistance pattern of all positive cultures from the surgeries. All patients included in the study were evaluated according to the 2018-Musculoskeletal Infection Society (MSIS) criteria. In total, 34 patients fulfilled the 2018-MSIS criteria for infection after resection arthroplasty.

Moreover, CRP values of all the patients were recorded one week before and seven weeks after resection arthroplasty. CRP values were obtained 1–3 days preoperatively and on days 3 (±1), 10 (±1), 17 (±1), 24 (±1), 31 (±1), 38 (±1) and 45 (±1), postoperatively. We evaluated NPWT after resection arthroplasty procedures in patients with persistent wound drainage. Patients with persistent incisional drainage and delayed wound healing received NPWT.

The empiric antibiotic administration during the treatment period was based on our institutional guidelines, and when the microbiological results of a positive culture were available, the regimen was altered in accordance with our infectious disease specialist. All patients received intravenous antibiotics followed by chronic oral suppression for a minimum of 12 weeks. All patients were treated with a minimum combination of two antimicrobial agents. The initial empiric treatment for a suspected PJI was cefuroxim and moxifloxacin.

The primary endpoint of this study was defined as success or failure of a resection arthroplasty. Success was defined as successful reimplantation, resolution of all clinical signs and symptoms of infection and no microbiological relapse. Failure was defined as (i) persistent resection arthroplasty with no clinical improvement, or (ii) worsening/relapse of current infection, or (iii) new signs/symptoms of infection requiring either a change/addition of antibiotic therapy, or (iv) the requirement of additional surgical procedures.

All parameters were compared between the failure and the success groups. The minimum follow up was 12 months, by using the latest medical records or by performing follow-up calls. Patients with incomplete data sets or with follow up of less than 12 months were excluded.

### 2.3. Microbiological Analysis

At least two tissue samples (median: 5, range 2–11) were collected intraoperatively, and all explanted devices were further processed in sonication containers [29]. The container was sonicated in an ultrasound bath for 1 min and vortexed for 30 s. Tissue samples and aliquots (0.1 mL) of the sonication fluid were plated in aerobic and anaerobic sheep blood agar as well as in chocolate agar plates. Cultures were incubated at 35 ± 1 °C for 14 days. Identification of bacterial species as well as of *Candida* spp. and antimicrobial susceptibility testing were performed. In addition, interpretative criteria for antibiotic resistance established by EUCAST were used [29].

### 2.4. Statistical Analysis

We employed descriptive statistics, including mean (M), standard deviation (SD), median (Md), range by minimum (min), maximum (max) and percentage to present the characteristics of study parameters. Comparing study groups, Mann–Whitney U testing for metric variables was conducted. For nominally scaled variables, we used chi-squared testing based on crosstabs. To indicate the position of the expected value in the population, 95% confidence intervals (CIs) were calculated.

The Kaplan–Meier survival method was used to assess the reimplantation rate or the revision-free-survival interval. A log-rank test was performed in each Kaplan–Meier survival analysis.

For CRP values, the receiver operating characteristic (ROC) and the area under the curve were calculated. *p*-values < 0.05 were considered statistically significant.

To mimic some particular characteristics of a randomized controlled trial, propensity score matching (PS matching) was used to match the NPWT group and the non-NPWT group and CRP values from the failure group and the success group, by using defined baseline covariates [30]. BMI, age, SSI and CCI were the baseline covariates [27,28]. A z-score was created for each of these covariates to create one total z-score including all covariates. The most similar z-scores between two groups were matched. Data analyses were performed using IBM SPSS^®^ version 25 and GraphPad Prism 8 (accessed on 9 August 2021).

## 3. Results

### 3.1. Patient Demographics and Factors Associated with a Failed Two-Stage Protocol

In total, 61 patients, 42 female (68.9%) and 19 male (31.1%), were included in this study. The mean age of patients was 68.51 (±10.52) years, with a mean body mass index of 28.47 (±6.04). There were 37/61 (60.7%) patients in the success group and 20/61 (32.8%) patients in the failure group. Overall, the median infection-free prosthesis survivorship was 2.5 (range 1–9.5) years in the success group. However, four patients died within the follow-up period at days 12, 73, 88 and 126 after resection arthroplasty.

In 37/61 (60.7%) patients, a successful reimplantation was performed; 10/61 (16.4%) had a re-revision procedure following reimplantation (9/10 debridement, 1/10 spacer implantation), 10/20 (16.4%) had a permanent Girdlestone situation, and 4/61 (6.6%) died within the follow-up period.

There were significantly higher CCI (*p* = 0.002), more elderly people (*p* = 0.018) and a higher number of operations prior to the resection arthroplasty (*p* = 0.022) in the failure group when compared to the success group. Moreover, the number of resection arthroplasties performed after revision surgery was higher in the failure group (*p* = 0.034) than in the success group. The median prosthesis-free interval in the success group was 84 (from 45 to 638) days. The demographic and other baseline characteristics of the patients included in this group are shown in Table 1.

Kaplan–Meier survivorship curves were generated using the revision-free survival, microbiological results, CCI and infected primary or infected revision THA.

Figure 1a shows the revision-free survival rate at 12 months. The revision-free survival was 80.83% in patients who underwent a reimplantation (*n* = 37) and 51.48% in patients with a permanent Girdlestone situation (*n* = 20). There was significantly higher revision-free survival in the reimplantation group compared with that in the permanent Girdlestone group (log-rank test, *p* = 0.003).

Moreover, microbiological results showed that the reimplantation rate was 55.57% for positive cultures (*n* = 41) and 61.04% for negative cultures (*n* = 16) after resection arthroplasty within the follow-up period (Figure 1b). There was no significant distribution when comparing the reimplantation rate between patients with negative cultures and positive cultures (log-rank test, *p* = 0.741).

The reimplantation rate was 77.78% in patients with CCI 0–3 (*n* = 27) and 41.77% in patients with CCI ≥ 4 (*n* = 30) after resection arthroplasty within the follow-up period (Figure 1c). There was a significantly high reimplantation rate in patients with a CCI lower than 4 (log-rank test, *p* = 0.033).

In addition, the reimplantation rate was 71.01% in patients when the resection arthroplasty was performed after primary THA and 51.28% when the resection arthroplasty was performed after revision THA within the follow-up period (Figure 1d). There was a significantly higher reimplantation rate in patients with resection arthroplasty after primary THA compared to that in those who underwent resection arthroplasties after revision THA (log-rank test, *p* = 0.021).

### 3.2. Microbiological Results

Microbiological cultures from 104 procedures were analyzed: 57/104 (54.8%) during the resection arthroplasties and 47/104 (45.2%) during reimplantations or re-revision surgeries. There were 50/104 (48.1%) procedures that resulted in a positive culture. In total, 41/57 (71.9%) resection arthroplasties (success group: 26/37 (70.3%); failure group: 15/20 (75.0%); *p* = 0.704) and 9/47 (19.1%) reimplantation/re-revision procedures resulted in positive cultures (success group: 5/37 (13.5%); failure group: 4/10 (40.0%); *p* = 0.059).

There were 7/47 (14.9%) patients with positive cultures at resection and reimplantation/re-revision procedures (success group: 3/37 (8.1%); failure group: 4/10 (40.0%); *p* = 0.012).

Microbiological changes occurred in all cases, 3/3 (100%), in the success group and in 3/4 (75.0%) in the failure group (*p* = 0.350).

There were 26/47 (55.3%) patients with positive culture results at first stage and negative culture results at reimplantation/re-revision (success group: 23/37 (62.2%); failure group: 3/10 (30.0%); *p* = 0.070). The microbiological results and the microbiological changes are shown in Table 2.

Overall, we detected 64 microorganisms in 43 positive culture procedures, including 31 different species. There were 81.3% (52/64) Gram-positive microorganisms, 14.1% (9/64) Gram-negative microorganisms, and 4.7% (3/64) *Candida* species. *Staphylococcus epidermidis* (29.7% (19/64)) and *Staphylococcus aureus* (9.4% (6/64)) were the most common microorganisms.

Polymicrobial infections were found in 11/104 (10.6%) procedures (success group, 7/37 (26.9%); failure group, 4/20 (25.0%)). There were 26 microorganisms detected in the 11 polymicrobial resection arthroplasty procedures. *Staphylococcus epidermidis* 9/26 (34.6%) and *Staphylococcus haemolyticus* 2/26 (7.7%) and *Enterococcus faecalis* 2/26 (7.7%) were the most relevant among the polymicrobial infections. The microbiological results are shown in Table 3.

### 3.3. CRP Values

In total, 1089 CRP values were recorded from 61 patients. For the final analysis, 467 CRP values were collected during our predefined time range. There were 120 (25.7%) CRP values taken 1–3 days prior to the resection arthroplasty, 86 (18.4%) on day 3 (±1), 65 (13.9%) on day 10 (±1), 47 (10.1%) on day 17 (±1), 46 (9.9%) on day 24 (±1), 29 (6.2%) on day 31 (±1), 42 (8.9%) on day 38 (±1) and 32 (6.9%) on day 45 (±1) postoperatively. To compare the CRP values between the success group and the failure group, a 1:1 PS matching method, using the predefined covariates was performed in order to minimize the heterogeneous covariate influence on the outcome. There was no significant difference in CRP values between the matched failure group and the success group preoperatively on days 3 (±1), 10 (±1), 38 (±1) and 45 (±1) postoperatively. However, there were significantly higher CRP values in the failure group on days 17 (±1), 24 (±1), 31 (±1) postoperatively (*p* ≤ 0.05). The longitudinal CRP trend and the AUC after ROC are depicted in Table 4 and Figure 2.

### 3.4. NPWT Results

Out of the 61 patients, 12 (19.7%) patients required NPWT due to persistent wound drainage and 49 (80.3%) received standard dressings. Patients received NPWT after a median of 5 (3; 12) days postoperatively. There were significantly more patients in the failure group 9/20 (45.0%) than in the success group 3/37 (8.1%) who required NPWT (*p* = 0.004). A 2:1 PS analysis method was performed in order to compare the patients requiring NPWT with those who did not need NPWT. This approach was followed to minimize the heterogeneity on the patients’ outcome (Figure 3). In total, 3/12 (25%) patients with NPWT were successfully reimplanted, whereas 15/24 (62.5%) patients without NPWT were reimplanted, within the follow-up period.

## 4. Discussion

This study provides insights about causes and patients risk factors for renewed failure and infection persistence or eradication in a two-stage procedure with resection arthroplasty. Positive cultures from re-revision surgeries after resection arthroplasty are significantly highly associated with a failed two-stage protocol outcome. Moreover, persistently high CRP values, high CCI, re-revisions after the initial revision THA and the requirement of NPWT are potential risk factors associated with the failure of a two-stage protocol.

Comorbidity indices can be easily obtained pre-operatively for all patients and may help to assess their individual risks [31]. Our study also showed that CCI > 4 was a risk factor associated with a failed two-stage protocol. An increased rate of persisting PJI after revision surgery for initial PJI has been described in the literature [32]. In our analysis, patients with resection arthroplasty after revision THA showed a lower reimplantation rate compared to those who underwent resection arthroplasty after primary THA. This could be partially justified by the high rates of persistent infections in revision THA.

Moreover, we found a higher re-revision rate in patients with permanent Girdlestone (49.52%) compared to patients submitted to a reimplantation (19.17%) one year after resection arthroplasty. The study from Leitner et al. described a 65% revision rate after 10 years from the Girdlestone resection arthroplasties [33]. A permanent Girdlestone situation does not prevent further re-revisions.

Infection eradication is important for a successful outcome of a two-stage protocol [34]. Proper pathogen identification and determination of the antibiotic susceptibility profiling are important tools to improve PJI treatment [35]. In our study, a positive culture resulting from a resection arthroplasty and a positive culture during reimplantation/re-revision often resulted in failure. Eradication of the pathogen during the second-stage procedure is a factor for treatment success.

Changes in the microbiological spectrum are commonly observed between the first and second stages and had to be considered in the antimicrobial treatment of PJI. In the study from Frank at al. the microbiological spectrum changes occurred in 80.0% of different surgical procedures [36]. In our study, microbiological spectrum changes occurred in 85.7% of the cases, both in the failure and in the treatment group. Accurate microbiological results are necessary for a successful infection eradication.

A recent study showed that patients with recurrent infections had increased CRP levels [37]. Moreover, patients with knee and hip PJIs showed higher CRP values in the septic group than in the aseptic group [38]. Overall, CRP evaluation is cheap, accessible, sensitive (from 68% to 94%) and more accurate than other serum parameters, such as white blood cell count, percentage of neutrophils or neutrophils to lymphocytes ratio [38,39,40]. In this study, we also found a significantly high CRP level in the failure group after the second week of resection arthroplasty. The diagnostic accuracy of synovial fluid parameters is higher than that of serum [11,41]. However, a synovial fluid aspiration is not always possible in hip PJIs. Persistently elevated serum CRP levels after resection arthroplasty may predict recurrent infections. Other causes of increased CRP levels should be excluded and put in context with additional clinical factors.

In most studies, NPWT was applied on non-leaking, non-delayed healing, primary hip arthroplasty wounds. In the study from Fröschen et al., a 36% success rate was reported after NPWT in PJI wounds [42]. In this study, a success rate of 25% after NPWT was found [23,27]. The requirement of NPWT is an indication for repeat debridement as a sign of persistent PJI just for medically unfit patients. Such patients cannot undergo a short-term re-revision and would benefit from a decreased edema, improved removal of exudate and increased blood and lymphatic flow when having NPWT treatment.

Patients who are at a high risk of resection arthroplasty failure in a two-stage protocol, should be evaluated for all the above risk factors.

There are some limitations in this study such as the retrospective nature of the study design with all its disadvantages and the short follow-up time. Additionally, the study did not include an evaluation of functional outcome as it was discussed previously [43,44,45]. Other limitations are the restricted number of patients and dependence on medical records prepared by different clinicians during a period of 10 years. Due to the small sample size, multivariate and additional univariate analyses were not conducted. Nevertheless, this matched retrospective study represents the largest single-center observation for risk factors in resection arthroplasty as the first stage of a two-stage procedure.

Moreover, the decision to perform a resection arthroplasty was based on a diagnosis of local bone deficiency made by the surgeon. Criteria to determine the type of procedure a patient should be submitted to were mostly based on the clinicians’ judgment, which differs according to the surgeons and their experience. There were no standard criteria to determine whether a resection arthroplasty procedure should be performed. Further prospective and randomized clinical trials are necessary to confirm these preliminary results. Risk factors in this study are non-comprehensive. Additional factors, such as surgical techniques, surgeon’s experience and the duration of the operation, were not analyzed in this study. In conclusion, all the risk factors addressed in this study seem to be equally important in helping to define the outcome of resection arthroplasty as a first-stage procedure in patients with severe hip infections. High CCI, resection arthroplasty after infected revision THA, high number of previous surgeries, the requirement of NPWT, continuous high serum CRP levels and positive culture results at reimplantation/re-revision arthroplasty have been identified. These risk factors need to be evaluated when counseling these patients.

## Figures and Tables

**Figure 1 jcm-10-05180-f001:**
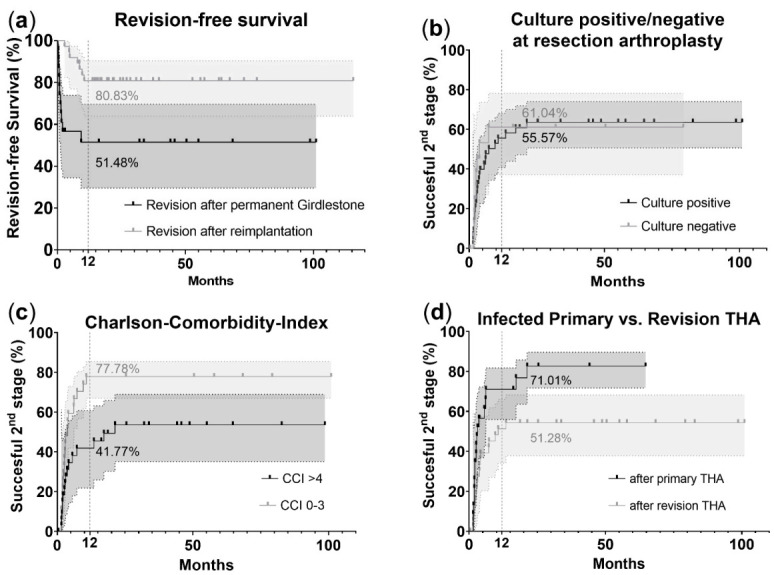
Kaplan–Meier curves illustrating the likelihood for (**a**) revision-free survival, (**b**) positive/negative cultures at resection arthroplasties, (**c**) Charlson comorbidity index (CCI) and (**d**) reimplantation after primary and after revision total hip arthroplasties. The Kaplan–Meier curves show plotting outcomes for two different groups in each analysis and its 95% CI interval. *p* values reflect the probability of a failed outcome in patients with CCI > 4 and resection arthroplasties after revision surgeries, as well as a high re-revision rate in patients with a permanent Girdlestone situation. Total hip arthroplasties (THAs).

**Figure 2 jcm-10-05180-f002:**
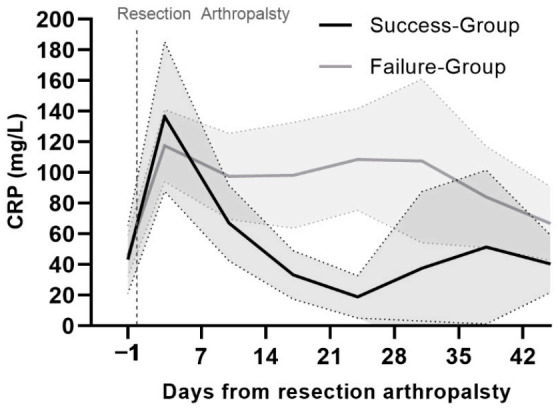
Longitudinal CRP trend and the area under the curve (AUC) after receiver operating characteristic (ROC) for successful (patients with reimplantation after resection arthroplasties) and failed resection arthroplasties (mean 95% CI interval).

**Figure 3 jcm-10-05180-f003:**
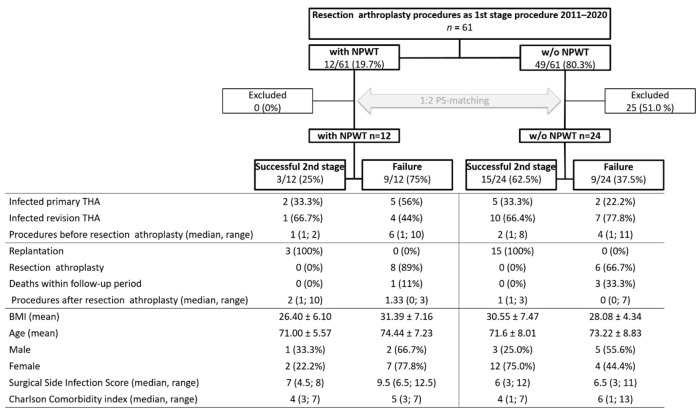
Comparison of risk factors between patients who received negative-pressure wound therapy (NPWT) and patients who did not receive NPWT (w/o) group before and after propensity-score (PS) matching.

**Table 1 jcm-10-05180-t001:** Comparison of demographics and risk factors between successful two-stage procedures (success group) and failed two-stage procedures (failure group).

Parameter	Success Group *n* = 37	Failure Group *n* = 24	*p*-Value
Age	65.81 (±11.05)	72.67 (±8.24)	0.018 *
Gender: female	27 (73.0%)	15 (62.5%)	0.358
male	10 (27.0%)	9 (37.5%)	
BMI	27.92 (±6.45)	29.33 (±5.36)	0.372
SSI	6.50 (±2.38)	7.54 (±2.79)	0.126
CCI	3.24 (±2.02)	5.46 (±2.89)	0.002 *
Res. arthroplasty after primary THA	18 (48.6%)	5 (20.8%)	0.034 *
Res. arthroplasty after revision THA	19 (51.4%)	19 (79.2%)	
Operation prior to res. arthroplasty	2.30 (±1.79)	3.88 (±2.88)	0.022 *
Time between prim. implantation and res. arthroplasty (months)	89.69 (±78.80)	107.22 (±107.94)	0.763
Operation after res. arthroplasty	1.51 (±0.96)	1.25 (±1.87)	0.472
Prosthesis-free interval	84 (45; 638)	-	
NPWT required after res. arthroplasty	3 (8.1%)	9 (37.5%)	0.005 *

BMI, body mass index; SSI, surgical site infection; CCI, Charlson comorbidity index; NPWT, negative-pressure wound therapy; Res., resection; THA, total hip arthroplasty; * *p* ≤ 0.05.

**Table 2 jcm-10-05180-t002:** Comparison of microbiological results and microbiological changes between successful two-stage procedures (success group) and failed two-stage procedures (failure group).

		Success Group		Failure Group	
	Total	Res. Arthroplasty *n* = 37	Reimplantation 37 (100.0%)	Res. Arthroplasty *n* = 20	Re-Rev. 10 (50.0%)
Cult. neg.	54	11 (29.7%)	32 (86.5%)	5 (25.0%)	6 (60.0%)
Cult. pos.	50	26 (70.3%)	5 (13.5%)	15 (75.0%)	4 (40.0%)
Monomicrobial	39	19 (73.1%)	5 (100%)	12 (80.0%)	3 (75.0%)
Polymicrobial	11	7 (26.9%)	0	3 (20.0%)	1 (25.0%)
Cult. Neg. → Cult. Pos.	2	2/37 (5.4%)		0/10 (0.0%)	
Cult. Neg. → Cult. Neg.	12	9/37 (24.3%)		3/10 (30.0%)	
Cult. Pos. → Cult. Neg.	26	23/37 (62.2%)		3/10 (30.0%)	
Cult. Pos. → Cult. Pos.	7	3/37 (8.1%) *		4/10 (40.0%) *	
Spectrum changed	6	3/3 (100%)		3/4 (75.0%)	

Res, resection; re-rev, re-revision; cult, culture; Pos, positive; Neg, negative. * *p* ≤ 0.05.

**Table 3 jcm-10-05180-t003:** Microorganisms identified during resection arthroplasties, reimplantations and re-revision surgeries in successful two-stage procedures (success group) and failed two-stage procedures (failure group).

		Success Group		Failure Group	
	Total	Res. Arthroplasty *n* = 37	Reimplantation 37 (100.0%)	Res. Arthroplasty *n* = 20	Re-Rev. 10 (50.0%)
Total microorganisms	64	36	5	19	4
*Staphylococcus epidermidis*	19 (29.7%)	9 (25.0%)	3 (60.0%)	6 (31.6%)	1 (25.0%)
MRSE	14	6	2	5	1
MSSE	5	3	1	1	-
*Staphylococcus aureus* (MSSA)	6 (9.4%)	2 (5.6%)	-	4 (21.1%)	-
*Cutibacterium* spp.	6 (9.4%)	3 (8.3%)	1 (20.0%)	1 (5.3%)	1 (25.0%)
*Cutibacterium acnes*	5	2	1	1	1
*Cutibacterium avidum*	1	1	-	-	-
*Staphylococcus haemolyticus*	4 (6.3%)	3 (8.3%)	-	1 (5.3%)	-
*Enterococcus faecalis*	4 (6.3%)	3 (8.3%)	-	1 (5.3%)	-
Viridans group *Streptococci*	4 (6.3%)	1 (2.8%)	1 (20.0%)	2 (10.6%)	-
Other Gram-positive bacteria	9 (14.1%)	8 (22.2%)	-	-	1 (25.0%)
*Beta hemolytic Streptococci*	1	1	-	-	-
*Staphylococcus lugdunensis*	1	1	-	-	-
*Staphylococcus capitis*	1	-	-	-	1
*Staphylococcus hominis*	1	1	-	-	-
*Staphylococcus canis*	1	1	-	-	-
*Clostridium perfringens*	1	1	-	-	-
*Micrococcus luteus*	1	1	-	-	-
*Kocuria rhizophila*	1	1	-	-	-
*Lactobacillus* spp.	1	1	-	-	-
*Gram-negative bacteria*	9 (14.1%)	6 (16.6%)	-	3 (15.9%)	-
*Enterobacter cloacae*	1	1	-	-	-
*Escherichia coli*	2	2	-	-	-
*Citrobacter koseri*	2	1	-	1	-
*Pseudomonas aeruginosa*	2	1	-	1	-
*Moraxella osloensis*	1	1	-	-	-
*Bacteroides fragilis*	1	-	-	1	-
*Candida* spp.	3 (4.7%)	1 (2.8%)	-	1 (5.3%)	1 (25.0%)
*Candida parapsilosis*	1	1	-	-	-
*Candida albicans*	2	-	-	1	1

MRSE, methicillin-resistant *Staphylococcus epidermidis*; MSSE, methicillin-susceptible *Staphylococcus epidermidis*; MSSA, methicillin-susceptible *Staphylococcus aureus*.

**Table 4 jcm-10-05180-t004:** Comparison of C-reactive protein (CRP) values’ trend pre/postoperatively after resection arthroplasty between successful two-stage procedures (success group) and failed two-stage procedures (failure group).

	CRP Levels (mg/L)			
Parameter	Success Group	Failure Group	*p*-Value	AUC
CRP values total	57.59 (±81.25)	115.88 (±201.41)	0.003 *	0.596
2 (±1) days pre-OP	43.19 (±40.18)	45.84 (±53.94)	0.955	0.495
3rd (±1) day post-OP	136.6 (±123.47)	117.51 (±79.55)	0.986	0.499
10th (±1) day post-OP	67.16 (±55.81)	97.60 (±76.34)	0.118	0.627
17th (±1) day post-OP	33.17 (±27.63)	98.22 (±88.90)	0.004 *	0.777
24th (±1) day post-OP	18.95 (±22.99)	108.54 (±90.46)	0.001 *	0.813
31st (±1) day post-OP	37.56 (±59.81)	107.59 (±100.06)	0.032 *	0.773
38th (±1) day post-OP	51.41 (±74.75)	84.16 (±79.98)	0.071	0.691
45th (±1) day post-OP	40.42 (±27.79)	66.74 (±39.79)	0.093	0.703

* *p* ≤ 0.05. Area under the curve (AUC) after receiver operating characteristics analysis.

## Data Availability

Data available on request due to privacy restrictions. The data presented in this study are available on request from the corresponding author. The data are not publicly available due to GDPR.

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
