# Peer review of "Analysis of Failed Two-Stage Procedures with Resection Arthroplasty as the First Stage in Periprosthetic Hip Joint Infections"

_jcm, 2021, doi:10.3390/jcm10215180_

Round 1

Reviewer 1 Report

The authors present a paper on a important topic. They clearly have substantial experience treating PJI of the hip as they treat around 50 cases per year.

However, major concerns need to be addressed. Writing and presentation of results need drastic improvement. There are  way too many "risk factors" explored for a population of only 61 patients. Also, biased risk factors, such as the use of NPWT are explored, which in my opinion should be seen as a early sign of treatment failure rather than as a risk factor on its own. Definitions of PJI, culture negative PJI, failed resection arthroplasty, etc need to be provided.

Page 1 line 1: the title is confusing. Now it seems the surgeons were not able to explant the infected THR.

Page 1 line 12: probably better to use "factors associated with"  instead of risk factors, given this is a retrospective study

Page 1 line 16-17: The definition of success / failure needs to be clarified. What about patients who underwent a 2nd stage, but infection persisted?

Page 1 line 24-26: not sure why evaluation of microbiological results, etc would lead to better outcomes in these kind of patients? Conclusion needs to be improved.

Page 1 line 34: not sure what this means. If you had knowledge of these predictive risk factors, how would it change your treatment? All the "risk factors" found in the study are non-modifiable, age, CRP, previous operations.

Page 1 line 38-40: what else can an unsuccesful 2 stage exchange be related to if not unsuccesful eradication of the infection?

Page 2 line 52: this is not entirely true, for example: Wang Q, Goswami K, Kuo FC, Xu C, Tan TL, Parvizi J. Two-Stage Exchange Arthroplasty for Periprosthetic Joint Infection: The Rate and Reason for the Attrition After the First Stage. J Arthroplasty 34(11): 2749, 2019

Page 2 materials and methods: what are the authors indications for 1 stage, 2 stage with and without spacer? What are the success rates of these other procedures?

Page 2 line 82-83: for how long?

Page 3 line 92: how many?

Page 4 line 149: what is the definition of  negative PJI? Moreover, which definition of PJI has been used?

Page 4 lines 141-158: difficult to read - presentation needs to be improved

Page 5 lines 164-178: same

Page 6 line 182: what is meant by "we detected 64 micro-organisms"? 64 positive samples? 64 operations with positieve samples?

Page 9 line 232: there is no way that this paper provides detailed information on the causes of failure after resection arthroplasty. Perhaps some associated factors.

Reviewer 2 Report

This is a well written manuscript on an important topic that will be of significant interest to the readers in arthroplasty. However, minor revisions to grammar and spelling are required before it is acceptable for publication.

Please see attached pdf for suggested grammar and spelling changes.

In the conclusion paragraph, you are saying there is no single predictive risk factor superior to others. But what are the risk factors that increase the risk of failure after resection? List them and their effect on outcome. What microbiological results affect outcome? What serum values? Which patient demographics? What wound status? Summarize whether or not each of these risk factors you listed are predictive of failure. 

Reviewer 3 Report

Methods: please detail technique. "resection arthroplasty" might be confusing. Please detail and specify.

Please add citations for sonication (eg sambri et al, Clin Orthop Relat Res. 2018

Also, were cement spacer sonicated at the time of 2nd stage? (Sambri et al J Microbiol Methods. 2019)

"Culture -negative PJIs were defined as one for which cultures of intraoperative tissue samples did not isolate an organism. "This might be true if other MSIS criteria were matched.

Follow up is very short as to verify main endpoint. This must be acknowledged as a major limitation. Also, the series is very small.

Author Response

Dear Reviewer,

Enclosed please find our revised version of manuscript jcm-1363013 entitled “Analysis of failed Two Stage Procedures with Resection Arthroplasty as First-Stage in Periprosthetic Hip Joint Infections” for the journal of clinical medicine.

We would like to thank the Reviewer for his excellent comments, and believe that our manuscript has been significantly improved by incorporating these comments. Each comment raised by the reviewers was addressed and we hope that our revised manuscript is now suitable for publication. 

Thank you for your consideration. 

Round 2

Reviewer 1 Report

NA

Author Response

(The authors gave the same response as above.)
